# Maintaining, Managing, and Tele-Monitoring a Nutritionally Adequate Mediterranean Gluten-Free Diet and Proper Lifestyle in Adult Patients

Alice Scricciolo [1], Karla A. Bascuñán [1,2], Magdalena Araya [3], David S. Sanders [4], Nick Trott [4], Luca Elli [1], Maria Teresa Bardella [1], Luisa Doneda [5], Vincenza Lombardo [1], Nicoletta Nandi [1], Maurizio Vecchi [6,7] and Leda Roncoroni [1,5,*]

1   Center for Prevention and Diagnosis of Celiac Disease, Gastroenterology and Endoscopy Unit, Fondazione IRCCS Ca' Granda Ospedale Maggiore Policlinico, 20122 Milan, Italy; alice.scricciolo@policlinico.mi.it (A.S.); kbascunan@uchile.cl (K.A.B.); dottorlucaelli@gmail.com (L.E.); mariateresa.bardella@yahoo.com (M.T.B.); vincenza.lombardo@policlinico.mi.it (V.L.); nicoletta.nandi@unimi.it (N.N.)
2   Department of Nutrition, Faculty of Medicine, University of Chile, Santiago 8380453, Chile
3   Institute of Nutrition and Food Technology (INTA), University of Chile, Santiago 7830490, Chile; maraya@inta.uchile.cl
4   Academic Unit of Gastroenterology, Royal Hallamshire Hospital, Sheffield S10 2JF, UK; david.sanders1@nhs.net (D.S.S.); nick.trott@nhs.net (N.T.)
5   Department of Biomedical, Surgical and Dental Sciences, University of Milan, 20100 Milan, Italy; luisa.doneda@unimi.it
6   Department of Pathophysiology and Transplantation, University of Milan, 20100 Milan, Italy; maurizio.vecchi@unimi.it
7   General Surgery Unit, Fondazione IRCCS Ca' Granda Ospedale Maggiore Policlinico, 20100 Milan, Italy
*   Correspondence: leda.roncoroni@unimi.it; Tel.: +39-025-503-3384

**Abstract:** The gluten-free diet (GFD) is a restrictive diet. In many cases, it must be permanent and strict, and it may be associated with both nutritional deficiencies and excesses, which can be prevented by following a healthy, natural Mediterranean GFD (Med-GFD). In this paper, we describe the importance of the Mediterranean diet, the correct intake of vitamins and minerals, and how they may play an important protective role against chronic or degenerative conditions. Herewith, we analyze different aspects that influence the ability to maintain a correct and balanced Med-GFD, which may contribute to the health status of patients, including a conscious use of gluten-free products to maintain a healthy lifestyle. Monitoring the Med-GFD remains a pivotal issue: to evaluate the presence of gluten peptides in urine, it could be important to introduce point-of-care testing, an efficient method for GFD self-monitoring (immunochromatographic technique), together with online nutritional questionnaires. Indeed, medical care via telemedicine can provide practical indications aimed at supporting patients and doctors. A natural Med-GFD can ensure the correct intake of nutrients and could be important for patients affected by gluten-related disorders, helping them to maintain a correct and healthy lifestyle.

**Keywords:** gluten-free diet; Mediterranean diet; telemedicine; telehealth; televisits; nutrition; gluten detection test

## 1. Introduction

Currently, the gluten-free diet (GFD) is recommended for gluten-related disorders with different targets and modalities; in fact, the GFD is usually administered temporarily in non-celiac gluten sensibility, while a strict and life-long adherence is required for celiac disease (CD) [1,2]. Being a restrictive diet, the GFD usually reduces the quality of life of patients, and in some cases, it could lead to nutritional deficiencies or imbalances. To support an adequate nutritional status, the GFD can be further improved with foods from the Mediterranean basin, ensuring a correct micronutrient intake. This diet is called the

Mediterranean GFD (Med-GFD). The Mediterranean diet is a healthy eating plan, inspired by the eating habits of the people who live in the Mediterranean basin. It varies by region and country and includes foods derived from plant sources such as fruits and vegetables, whole grains, pulses, and olive oil, as well as fish, moderate consumption of dairy products and eggs, and low consumption of red meats and other meat products [3].

The important change made with the Med-GFD is the type of cereal included in this diet, which consists of gluten-free cereals and pseudo-cereals, which are also typical of the Mediterranean diet.

In this context, there appears to be a need for adequate nutritional support for gluten-related disorders, as well as gastroenterological support, in both diagnostic and follow-up phases. The Med-GFD is adequate for both conditions, remembering that for naïve CD patients, it is important to avoid lactose and to adhere to a lactose-free diet (maintaining lactose-free products, aged cheeses, and UHT milk) during the first 6 months because of possible transient lactose intolerance. However, it may paint a clinical picture frequently characterized by a diarrheal habit; in this case, the intake of fiber could worsen symptoms and increase the frequency of evacuations. From this point of view, the interaction between the gastroenterologist and nutritionist appears to be pivotal for a customized and personalized therapy.

Furthermore, the recent COVID-19 (SARS-CoV2) pandemic has profoundly changed people's lifestyles and eating habits, illustrating the importance of a good immune system and an optimal health status to protect from severe forms of infection. Studies have confirmed that bad eating habits and unhealthy lifestyles can contribute to the development of an inflammatory state and a weak and vulnerable immune system. The pandemic has also evidenced the possibility of tele-monitoring patients on a Med-GFD [4–6].

Helping to preserve a nutritionally adequate Med-GFD, which, in turn, helps maintain a healthy intestinal microbiome and immune system, is the safest option to minimize the basal inflammatory state. In gluten-related disorders such as CD, increasing evidence has shown that the intestinal microbiota is modified [7,8], and in addition to this, the presence of gluten itself in the intestinal lumen induces functional changes [9]. Few studies have investigated adherence to the Med-GFD in the CD population. One study from the Mediterranean region demonstrated how a Med-GFD contributed to the correct nutritional status of CD patients without causing an overweight or obese condition, showing a similar body mass index (BMI) and body composition to those in non-celiac subjects [10].

For a Med-GFD to be adequate, balanced, and beneficial to patients, it has to be monitored and maintained throughout life.

## 2. Mediterranean Gluten-Free Diet and Nutritional Support

The diet in the Mediterranean basin consists of a traditionally familiar Mediterranean diet, which can be described as rich in plant-based foods such as fresh fruits and vegetables, pulses, nuts, herbs, olive oil, and whole wheat grains, with moderate consumption of dairy, poultry, eggs, and seafood, and low consumption of saturated and trans fats, animal-derived proteins, and refined sugars [11]. The strength of the Mediterranean diet is its richness in polyphenols and their immune-protective activity and anti-inflammatory role. As previously studied, the Mediterranean diet has a preventive and protective role against non-communicable diseases (NCDs) such as cardiovascular [12] and respiratory diseases, type 2 diabetes, obesity, and cancer [13], all of which are currently frequent in populations of both developed and developing countries. These pathologies have an increased mortality risk and are associated with a dysregulation of the immune system and a proinflammatory status. An adequate nutritional status is necessary to protect and preserve optimal immune system functions. Previous studies that have investigated the link between nutritional deficiencies and how the immune system works have shown that malnutrition leads to immune deterioration and, consequently, to an increased susceptibility to infection [14].

Following a restrictive diet such as the GFD often negatively impacts the intake of nutrients that are essential for processes that maintain life, and the outcome largely depends



on whether the disease is active or deactivated [15,16]. Newly diagnosed celiac patients and those who do not adhere to the GFD may be more affected; however, those who follow the GFD but consume manufactured gluten-free products in excess may also have negative consequences, and this may favor the development of conditions such as obesity or cardiovascular diseases [17,18]. We recently proposed the use of a food pyramid based on the Mediterranean diet, which emphasizes the addition of gluten-free components to the pyramid, and the resulting diet is not only gluten-free but also nutritionally satisfactory [19].

Maintaining a strict GFD seems to be one of the few effective ways available to patients who are affected by a gluten-related disorder to better prepare them to fight infections and face the difficulties of daily life caused by recurring infections. The GFD is effective, but being a restrictive diet, it requires supervision by a trained professional to monitor it because it is often low in protein, fiber, and vitamins and high in lipids, sugars, and salt [15]. Recent data suggest that the GFD modifies the intestinal microbiota signature, although the functional consequences of these changes remain unclear [20]. Celiac patients on the GFD are described to have a higher cardio-metabolic risk (obesity, dyslipidemia, insulin resistance, and metabolic syndrome) [21,22]. High consumption of foods that are rich in energy and fat has been interpreted as a way to compensate for dietary restrictions, both in adults and children with CD [18]. Efforts are currently being made to increase the levels of dietary fiber and decrease the sugar content in gluten-free foods [17], which undoubtedly contribute to the maintenance of a healthy GFD. Unfortunately, these improved gluten-free products are still not universally available, and many patients do not consume them because they tend to be more expensive.

Supplementation with micronutrients in order to restore stores deserves special attention in the GFD. Based on initial ferritin levels, Theethira et al. [16] recommended oral iron supplementation until stores are replenished, and intravenous iron in cases with severe iron deficiency anemia or oral intolerance. A high-iron diet (>20 mg per day) is considered a well-tolerated approach; when patients receive adequate education and instructions, it can effectively increase daily iron consumption [23]. Vitamins and minerals should be monitored and supplemented as needed until their stores are restored (Figure 1). Vitamin D is found naturally in many gluten-free foods, including various fish species such as salmon and herring, dairy products such as milk, yogurt, and cheeses (ricotta, stracchino, crescenza, mozzarella, Grana Padano DOP), eggs, beef, liver, and in smaller quantities in mushrooms [24–27]. The Recommended Levels of Nutrients and Energy Intakes (LARN) establish a recommended vitamin D daily intake of 15–20 μg [28]. For people who spend little to no time outside, ethnic minority groups with dark skin, and people who always cover their skin when outside, the recommendation is to take a daily supplement containing 10 μg all year round [29]. It is important that 20% of vitamin D needs are obtained from foods and 80% from sunlight exposure and sun blockers [30]. Folic acid and vitamin B12 must be checked periodically. Regarding vitamins A, C, and E, the LARN establish a daily intake of 600–700 μg, 85–105 mg, and 12–13 mg, respectively [28]. These vitamins are widely available in foods such as several red fruits and vegetables (apricots, melons, peaches, oranges, strawberries, peppers, carrots, pumpkins, tomatoes, and chili peppers), kiwi, broccoli, rocket salad, liver, eggs, dried fruit (almonds, nuts, and pistachios), and olive oil [27]. Ca and Zinc supplements should also be included in GFD long-term supervision programs. The dietary fiber recommendation is 25–35 g/day depending on age and gender, and dietary fiber intake must be encouraged while on the GFD by means of gluten-free cereals and pseudo-cereals, which are high in fiber content. A balanced gluten-free Mediterranean diet can provide the necessary amounts of micronutrients. Gluten-free cereals and pseudo-cereals, such as rice, whole wheat rice, quinoa, buckwheat, corn, teff, and fonio, are all good sources of micronutrients (including selenium and zinc) and important antioxidant molecules, and all of them are main components of the Mediterranean diet [19]. The Mediterranean diet pattern is a sustainable diet, which incorporates food security, and social-cultural, environmental, and economic welfare for future generations. The major challenge is that it is not consumed by the majority of the population in the Mediterranean

region, as well as its applicability to extra-Mediterranean regions. The composition of the Mediterranean diet and lifestyle is relatively standard; it is based on grains, olive oil, pulses, fresh fish and lean meats, and fruits and vegetables, minimizing the intake of processed foods and saturated fat. Each country preserves and promotes its culture and culinary traditions, and country-specific variations in dietary patterns are inevitable, but patients could incorporate these ingredients according to different cultures and traditions [31].

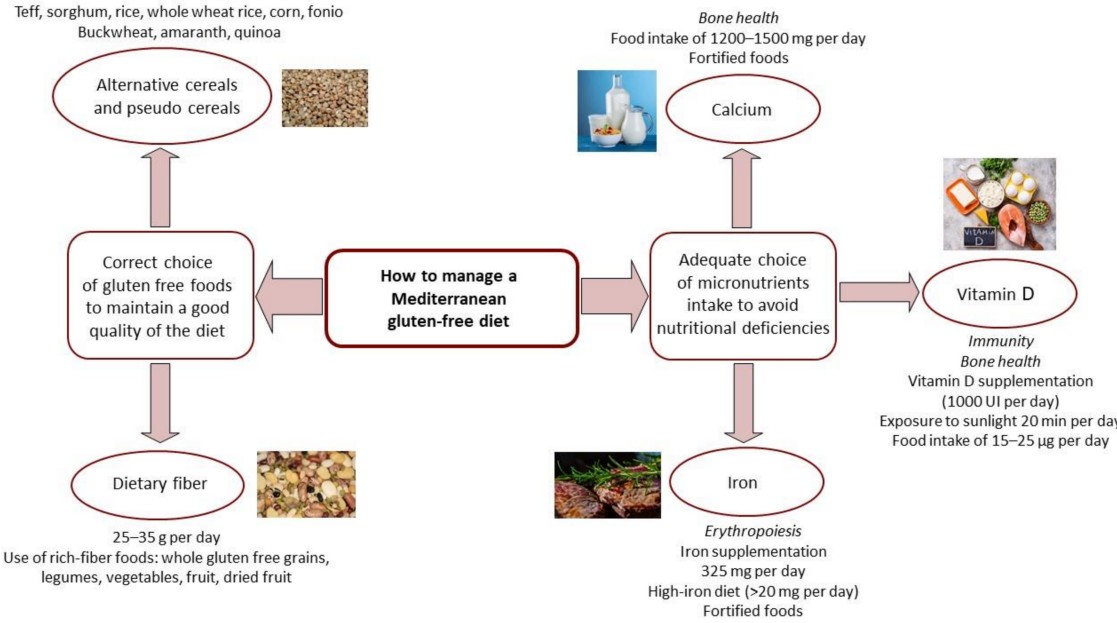

**Figure 1.** Recommendations for maintaining a healthy Mediterranean gluten-free diet.

However, adherence to a natural Med-GFD could be influenced by economic constraints. A strict adherence to a healthy dietary pattern is usually associated with consistent economic costs. Nevertheless, we should consider the fact that high adherence to a natural Med-GFD could be a protective factor against major diseases, reducing their incidence, prevalence, and related costs, as in the case of cardiovascular disease (CVD) [32].

Significant improvements in CVD-related costs and CVD reduction rates are evident when a correct Mediterranean diet is regularly followed. Specifically, CAD 41.9 million to CAD 2.5 billion in Canada and USD 1.0–62.8 billion in the United States were estimated as the total accumulated annual savings in economic costs [33,34].

### 3. Importance of a Healthy Lifestyle and a Natural Med-GFD

The substitution of industrial foods with naturally gluten-free foods is the most important factor, and nutritionists always advise avoiding industrially made gluten-free products while on a GFD. The consumption of gluten-free cereals and pseudo-cereals, such as buckwheat, quinoa, amaranth, rice, maize, teff, and fonio, should be preferred, as well as gluten-free whole grains and/or flours to cook home-made bread, cakes, biscuits, or pizza [19].

A healthy GFD includes milk and dairy products, especially partially skimmed milk, light cheeses, and yogurt, which are a great source of natural probiotics. Preferences should favor white meats (chicken, rabbit, and turkey) and fish (preferably codfish, seabass, sardines, and anchovies, while limiting swordfish, tuna, and cephalopods, such as squid and octopus) and limit the consumption of red meats (beef and pork).

Legumes are also called "pulses" (kidney beans, cannellini beans, Great Northern beans, navy beans, fava beans, cranberry beans, black beans, pinto beans, soybeans, black-eyed peas, chickpeas, and lentils). Legumes are nutritious, being rich in fiber, protein, carbohydrates, B vitamins, iron, copper, magnesium, manganese, zinc, and phosphorous.

Pulses are naturally low in fat as they are essentially free of saturated fats and cholesterol. One serving of legumes (equivalent to a half cup) provides proteins (20–45%) with essential amino acids, complex carbohydrates (±60%), and dietary fiber (5–37%) [35,36]. Legume intake is essential and should be paired with the consumption of cereals for a better nutritional balance. In general, all types of cereals (pasta, bread, rice, etc.) are low in the essential amino acid lysine but are rich in sulfur amino acids such as methionine. All types of legumes are poor in sulfur amino acids and rich in lysine. Therefore, their combination allows for the correct intake of all the essential amino acids useful for our physiology and ensures a complementary amino acid pattern.

In different societies, meals contain both cereals and legumes as a result of a traditional Mediterranean meal [36,37]. Fiber intake can be increased by consuming five portions of fresh fruits and vegetables per day; one should consume extra virgin olive oil (EVO) and limit salt, favoring the use of herbs and spices. All of the aforementioned foods that are naturally gluten free are readily available in supermarkets. The former COVID-19 lockdown has urged many people to cook meals themselves and thus has been essential in associating nutritional information with culinary skills. Cooking activities were rediscovered by families as a daily activity linked to a healthy lifestyle. The lockdown reinforced the conviviality during cooking and meals that is intrinsic to the lifestyle of Mediterranean countries.

However, the COVID-19 pandemic may not have had a significant influence on the dietary habits of all people, many of whom suffered from stress and anxiety during the lockdown. Studies reported poor food habits such as increasing snack and meal frequencies, comfort food intake, and alcohol intake, and a reduction in fresh food consumption. The lockdown and quarantines have had a negative effect on dietary patterns, accompanied by weight gain and higher BMIs [38,39].

Recent studies have demonstrated that the reduction in movement and activities during the lockdown caused a range of psychological disorders (depression, anxiety, stress, and sleeping problems), independent of geographic location. It has been estimated that during the pandemic lockdown, a decreased quantity and intensity of physical activity led to an increase in daily "sitting time" from 5.31 h/day before the confinement to 8.41 h/day during the confinement [40]. Recommendations from the American College of Sports Medicine indicate that exercise improves physical and mental health and/or fitness in all people, independent of their training habits. They indicate that flexibility, cardiorespiratory, resistance, and neuromotor exercises are recommended for healthy adults, regardless of age. Different approaches are recommended to overcome the current isolation situation, including home-based exercises, dancing to music, yoga, exergaming, aerobic activities, balance and flexibility exercises, and muscular strength and endurance training. Adults should exercise for at least 150 min (at a moderate intensity) and at least 75 min (at a vigorous intensity) weekly, divided into 5–7 sessions per week [41,42].

## 4. Self-Evaluation of GFD Correctness: Gluten Detection Tests

Noninvasive tools, including symptoms, serology (serum antibodies), dietary adherence questionnaires, and novel gluten immunogenic peptides, may detect ongoing villous atrophy rather than assess adherence to a GFD as in an invasive examination. The measurement of serum antibodies, in which a blood sample must be taken at a clinical laboratory or outpatient clinic, can also be determined today by using recently developed tools for self-assessment of serum antibody levels [43]. However, it must be kept in mind that we still need to settle the debate as to whether CD-related antibodies (TTG, EMA, and DGP) are reliable and accurate enough to assess adherence to a GFD, which is a critical step in evaluating the clinical course of CD during follow-up. On the other hand, nutritional follow-up is essential in these patients (Table 1). Today, new techniques using online questionnaires facilitate obtaining data both for a nutritional evaluation and adherence to a GFD. In addition, CD-specific health-related quality of life and food interviews allow the personalization of a GFD and the prescription of indications as needed. Furthermore, the introduction of point-of-care testing to evaluate the presence of gluten peptides in urine

could be an efficient method for self-monitoring a GFD [44]. Monitoring the GFD remains a pivotal issue. Even if certified alimentary products are available, special attention must be paid to gluten contamination. The nutritionist should support the patient in the correct food choices, explaining how to choose gluten-free packaged products, including pasta, bread, and biscuits, which can be included in the diet in association with a balanced macro- and micronutrient intake. Products with the certified gluten-free trademark must be chosen, paying attention to avoid foods with "traces" of gluten shown on the food label [45].

**Table 1.** Proposed steps to monitor the GFD via telemedicine.

| Proposed Steps | Healthcare Professionals | Assessments |
|---|---|---|
| Suspected gluten-related disorder | Physician | Clinical history<br>Family history<br>Assessment of symptoms and general health conditions<br>Indication for serum antibody analysis |
| Gluten-related disorder diagnosis | Physician | Education on gluten-related disorder<br>Supplementation to treat/prevent nutritional deficiencies<br>Referral to nutritionist<br>Search for deficiencies, complications, and other diagnoses |
| Gluten-related disorder follow-up | Physician | Assessment of general health status<br>Evaluation of persistent symptoms and signs<br>Evaluation and management of clinical findings<br>Request for serum antibodies within 6 months |
| Gluten-related disorder diagnosis | Nutritionist | Nutritional evaluation with self-reported body measures<br>Food education on gluten-free diet<br>Assessment and treatment of nutritional deficiencies<br>Activities, customs, and daily habits that influence diet<br>Food preferences and rejections<br>GFD eating plan design |
| Gluten-related disorder follow-up | Nutritionist | Nutritional evaluation with self-reported body measures<br>Adherence to the GFD (by questionnaire)<br>To assess specific health-related quality of life<br>(by questionnaire)<br>Viability and monitoring of the gluten-free eating plan<br>Evaluation of urinary gluten peptides |

Considering that gluten-free cooking may involve some difficulties, cooking at home could represent a risk of gluten contamination. Although professional nutritional telehealth support is an effective option today, patients themselves can successfully monitor and verify their eventual ingestion of gluten. We do not know if the diet needs to be as strict over time [46], but several on-the-spot techniques are now available which allow measuring urinary gluten peptides, fecal peptides, and blood antibodies; patients can easily control these at home when in doubt of gluten ingestion or when unexplained symptoms occur.

The use of multiple modalities to self-monitor the GFD could help, providing an important support to verify possible and unconscious gluten ingestion or contamination. The self-dosage test of gluten urinary peptides works via an immunochromatographic technique that uses the G12 monoclonal antibody (MoAb), which is able to detect GIP in urine within the previous 24–48 h [47–49]. Self-administered point-of-care tests are easy to use, convenient, cheap, trustworthy, and precise tools which can be used to indicate possible gluten contamination even during remote telehealth monitoring [50]. It is likely that these tools could increase the sensitivity to detect adherence problems. This is also a key aspect in the case of children with a gluten-related disorder, as their caregivers must be trained in using these alternatives.

## 5. Telemedicine to Monitor the GFD

Patients affected by gluten-related disorders were already under considerable pressure before the pandemic started due to regular clinical and dietetic follow-up sessions [1]. The medical care of these patients is traditionally provided during periodic visits to healthcare professionals, who evaluate overall health, nutritional status, and adherence to the GFD and measure CD-specific serum antibodies [51]. In the pandemic scenario, telemedicine as a support for patients has been favored, and data suggest that patients trust the combined gastroenterological and nutritional televisits [5,52]. It represents an excellent opportunity to transform our current vision of professional healthcare and reinvent it through available online technologies to welcome, include, and support patients who reported positive results with telemedicine during the pandemic [49–51]. However, telemedicine was shown to be an effective tool in nutritional monitoring of chronic diseases even before the pandemic [53–55]. The American Telemedicine Association defined telehealth as "technology-enabled health and care management and delivery systems that extend capacity and access" [56]. Once diagnosed with a gluten-related disorder, the GFD is initiated, and periodic monitoring of the presence and intensity of symptoms, adherence to the diet, the presence or appearance of other pathologies, and quality of life must be provided [57] (Table 1). A recent study assessing the effectiveness of online consultations in the follow-up of celiac patients showed that health problems were detected significantly more frequently using ad hoc online questionnaires, with no differences in the detection of growth problems and dietary disarrangements when compared with a group that was "traditionally" evaluated by face-to-face assessments [51]. The term telemedicine includes phone calls, televisits, and different remote point-of-care diagnostic tests to detect gluten intake and to monitor GFD adherence [5].

Adherence to the GFD could be evaluated through questionnaires, such as the Celiac Disease Adherence Test (CDAT) [57], which consists of seven questions with a score ranging from 7 to 35. Higher scores denote worse GFD adherence. It is a simple and sensitive tool that is easily administered.

Currently, new and different technologies help us to communicate via diversified tools such as computers and mobile phones, which are instruments through which it is possible to monitor diseases and comorbidities; in particular, with innovative and new mobile apps, it is possible to educate patients on how to self-monitor their pathologies and symptoms [58]. Undoubtedly, in the future, mobile apps for the GFD could be a method used to monitor patient adherence.

## 6. Conclusions

A natural Med-GFD guarantees the correct intake of macro- and micronutrients and could help patients affected by gluten-related disorders to maintain a correct healthy state in conjunction with a healthy lifestyle. Furthermore, gastroenterologists and nutritionists should be aware of its characteristics and use in daily clinical practice.

Together with the use of point-of-care tests to monitor the GFD and face-to-face visits, a novel system of disease monitoring centered on telemedicine and new technologies could support physicians in patient management.

**Author Contributions:** Conceptualization, A.S., L.E., K.A.B. and L.R; writing—original draft preparation, A.S., K.A.B., M.A., L.E., M.T.B. and L.R.; writing—reviewing, and editing, L.E., K.A.B., A.S., D.S.S., N.T., L.D., M.A., M.V., V.L., N.N. and L.R.; funding acquisition, L.E. All authors have read and agreed to the published version of the manuscript.

**Funding:** This research was funded by Fondazione IRCCS Ca' Granda and received grants from Italy's Ministry of Health and Lombardy's Regional Government Authority (Ministero della Salute e Regione Lombardia, grant number 2011-02348234). The Article Processing Charge was funded by Fondazione IRCCS Ca' Granda Ospedale Maggiore Policlinico and Università degli Studi di Milano, Milan, Italy.

**Institutional Review Board Statement:** Not applicable.

**Informed Consent Statement:** Not applicable.

**Data Availability Statement:** Not applicable.

**Conflicts of Interest:** The authors declare no conflict of interest.

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
