# Peer review of "Maintaining, Managing, and Tele-Monitoring a Nutritionally Adequate Mediterranean Gluten-Free Diet and Proper Lifestyle in Adult Patients"

_applsci, doi:10.3390/app12031578_

Round 1

Reviewer 1 Report

The present review aimed to describe the importance of gluten free diet based on Mediterranean model and of a proper lifestyle to improve celiac disease patient’s health status focusing the attention also on telemedicine role as a tool to monitor and support patients.

Although the topic is of great interest, major changes are required:

  • The Abstract is not in line with the main text that resulted not focused on the proposed aims. It would be useful to re-organize the work identifying specific sections each described more in details (i.e. Mediterranean gluten free diet, role of telemedicine…).
  • The use of a gluten-free Mediterranean diet is proposed to every celiac patients without any differentiation between newly diagnosed and subject in gluten free diet for more than 6 months. However, it should be underlined how to adapt gluten-free diet according to specific conditions.
  • In addition, gluten contamination should be described also in naturally-gluten free products going in deep on this topic.
  • Figure 1 is not complete and it suggest the use of some cereals that instead should be evaluated with attention by celiac patients (i.e. oats).
  • Talking about lifestyle, only one study in reported to describe the effect of the pandemic. Moreover, nothing is reported about changes in terms of health status (i.e. changes on body weight, BMI…).
  • The role of telemedicine in monitoring nutritional habits is interesting and easy-to-understand. However, further studies should be added.
  • English is not always adequate and understandable. Some words are not appropriate (i.e. “Alternative” cereals, Mediterranean “fashion”). Some sentences are not easily understandable (i.e. lines 156-160).
  • Paragraph 3 should be reorganized and re-write with attention. Some nutritional concept are not precise. Moreover, nutritional values of legumes are not correct. In addition, the title should be changed. No references on Mediterranean diet are presents.
  • Is this review focused on adults? It is not easy to understand this.
  • A careful check of references should be carried out due to the fact that there are cases in which there are no correspondence between reference in text and in list.

Reviewer 2 Report

Reviewer comments:

This manuscript presents an interesting review about the gluten-free diet, it consequence on the health of patient and the importance to following a Mediterranean diet. Apparently the topic is of potential interest of this journal.  There are some researches about the gluten-free diet but not in this form, this work presents a complete review on the effect of this diet on the patient health and the usefulness of tele-medicine to monitor patients. The review is clear, comprehensive and provides the necessary details about the field. Comparing to similar works, this review gives a deep knowledge about the gluten-free diet

Introduction:

Authors must describe briefly the characteristics of a Mediterranean diet, what is the difference between a Med-GFD and the author’s area to clarify the issue addressed in this review

Line 45: “GFD can be further improved ensure a Mediterranean “fashion”, the so called Mediterranean-GFD (Med-GFD)”

This sentence is not clear, authors must clarify

  1. Mediterranean gluten free diet and nutritional support

Line 62 until 76: authors provide information about Med-diet in general and not gluten-free diet particularly; they must clarify this point to locate the link between the Med-gluten-free diet and the subject of the review, if there are studies about gluten-free diet in the Mediterranean region, authors must cite it.

Line 161-168: authors provide information about legumes but they didn’t stay the complementarity with cereals and the ratio aiming to offer a better nutritional balance in amino-acids as reported by many authors

References: well cited, authors used a majority of recent references and also some old references which is good in review papers.

Round 2

Reviewer 1 Report

The manuscript has been improved by authors. However, few minor changes are still request:

  • Authors modified lines 59-64 by adding the avoidance of lactose. However, there are other scenarios to take into account for example fiber intake depending on intestinal villi damage. It is important to explain the pivotal role of a customized approach depending on patient’s characteristic and clinical conditions.
  • In lines 244-246, it should be better described how to improve and manage naturally gluten free product in order to avoid contaminations. Please try to re-write the sentence because it is not clear.
  • Figure 1 should be further modified in order to focus the attention on how to manage a Med-GFD diet. The figure could be divided in two different parts. The first one regarding the diet and so how to choose foods and how to improve the quality of the diet. The second part should be related to possible micronutrient deficiencies and possible supplementation explaining the importance of personalization depending on patient’s needs and characteristics (i.e. age, gender…)
  • Paragraph 3 has been improved; however, it should be better clarify the role of legumes and the concept of “complementarity”. Lines 188-194 are confusing in terms of essential amino acid concept. Please, better explain this part.
